# Parenting Styles, Mental Health, and Catastrophizing in Women with Chronic Pelvic Pain: A Case-Control Study

**DOI:** 10.3390/ijerph192013347

**Published:** 2022-10-16

**Authors:** Vânia Meira Siqueira-Campos, Lara Juliana Henrique Fernandes, José Miguel de Deus, Délio Marques Conde

**Affiliations:** Department of Gynecology and Obstetrics, School of Medicine, Federal University of Goiás, 74.605-050 Goiânia, Goiás, Brazil

**Keywords:** parenting styles, women’s health, anxiety, depression, coping skills, emotional regulation

## Abstract

Chronic pelvic pain (CPP) in women is a highly prevalent condition worldwide and requires multimodal treatment. Adverse childhood experiences have been associated with CPP in women, while allodynia and poor outcomes have been linked to pain catastrophizing in these patients. Pain perception has been associated with parenting style during childhood. The objective of this study was to investigate the association between parenting style, pain catastrophizing, anxiety, depression and CPP in women. A case–control study was conducted between May 2018 and August 2021 with 123 women with CPP and 123 pain-free controls. Questionnaires were used to collect participants’ data. The association between parenting style and CPP was assessed using multiple logistic regression, with odds ratios (OR) and 95% confidence intervals (CI) being calculated. The correlation between catastrophizing, pain intensity, pain duration, anxiety, depression, and parenting style in women with CPP was assessed using Spearman’s rank correlation coefficient (r). A higher frequency of low maternal care (60.7% versus 45.2%; *p* = 0.026), anxiety (79.7% versus 56.9%; *p* < 0.001), depression (73.2% versus 56.1%; *p* = 0.008) and physical violence (31.7% versus 14.6%; *p* = 0.003) was found in the CPP group compared to the controls. There was no association between parenting style and CPP in the adjusted analysis. A positive correlation was found between catastrophizing and pain intensity (r = 0.342; *p* < 0.001), anxiety (r = 0.271; *p* = 0.002), depression (r = 0.272; *p* = 0.002), and maternal overprotection (r = 0.185; *p* = 0.046). A negative correlation was found between anxiety and maternal (r = −0.184; *p* = 0.047) and paternal (r = −0.286; *p* = 0.006) care and between depression and maternal (r = −0.219; *p* = 0.018) and paternal (r = −0.234; *p* = 0.026) care. The present results suggest a significant but weak association of parenting style with pain catastrophizing, the mental health of women with CPP, and the way in which they experience pain.

## 1. Introduction

Chronic pelvic pain (CPP) is a highly prevalent condition in women worldwide, particularly during the reproductive years [1]. In around 35% of patients, no visible lesions are present [2]. CPP exerts negative effects on women that include poor mental health [3], poor quality of life, sexual dysfunction [4], relationship difficulties and impaired work capacity [5]. Since this condition requires complex treatment, and positive outcomes are limited, it has been associated with a heavy economic and social burden [6]. A multimodal, interdisciplinary approach in which the different physiological and pathological mechanisms involved are taken into consideration has been recommended to improve outcomes [7].

Pain is an inherently personal experience [8] involving a multidimensional process that encompasses biological, psychological [9], behavioral [10] and socio-environmental [11] factors, as well as different individual perceptions [12]. Consequently, biopsychosocial models involving self-regulatory systems [13,14,15] and the attachment theory [16] have been proposed in an attempt to explain the experience of pain, particularly persistent pain. These models take the phylogenetic evolution of the nervous system into consideration, with the focus being on human survival. In this respect, the basic needs of each individual are considered to be adaptation to the environment, bonding and security [13,17]. Through physiological (allostasis), emotional, behavioral, and social processes, the functions of different structures and neural circuits overlap to guarantee the internal stability of the body (homeostasis) [18], identify safety or danger signs, respond to conflict or fight situations, trigger the flight or freeze response, and create bonds of attachment and social coexistence [16,17]. Therefore, the nervous system works in conjunction with the endocrine and immune systems as a single functional entity, modulating the defense response against internal and external threats [19]. However, an impaired ability to assess risk can exacerbate or prolong defense responses, interfering with homeostasis and influencing the individual’s emotional states. This can generate body dysfunctions, altered pain signaling such as hyperalgesia and allodynia, and may interfere with social bonding and interactions [17].

Part of the human nervous system is structurally and/or functionally immature at birth [20], with the development of self-regulatory systems being increasingly modeled both by genetic aspects and by the socio-affective environment [11,20,21]. In this respect, human adults (in general the parents, or in their absence, other caregivers) play an essential role in acting as co-regulators of the developing child [17]. Caregivers are expected to provide a safe basis of affection and to stimulate progressive autonomy in the child [16]. On the other hand, children gradually construct positive or negative internal models of the self and of others based on the responsivity, consistency and sensitivity provided by their caregivers, allowing their needs, particularly when they are ill, afraid or in pain, to be satisfactorily met [16]. During these initial interactions, perceptions of parental care and overprotection [22] lead children to develop secure or insecure attachment patterns [23]. These patterns can have long term repercussions on their cognitive evaluation of threat, expression, and management of pain, coping strategies, emotional expression, and interpersonal relationships [24,25]. Ultimately, this leads to the finding that optimal parenting essentially consists of high care and low overprotection [22].

Dysfunctional parenting styles during childhood have been associated with chronic pain in adolescence [26] and in adulthood [24,27], with this association sometimes being mediated by depression [26,27,28]. Pain catastrophizing, a psychological tendency to negatively magnify the experience of pain, has been associated with insecure attachment patterns [29] as well as with parental support and the way pain symptoms, particularly in early childhood, are monitored [10]. Catastrophizing, as a pain coping response, has also been associated with a greater intensity of pain [30], allodynia [12] and poor outcomes [31] in women with CPP. Furthermore, adverse childhood experiences (ACEs) such as physical and emotional abuse have been associated with CPP in women [32].

Therefore, the current study aimed to analyze the frequency of parental bonding styles in women with CPP compared to a control group of pain-free women, to investigate the association between parental bonding and CPP, and to examine the association between pain catastrophizing and the intensity and duration of the pain, anxiety, depression, and parental bonding in CPP patients. Our initial hypotheses were that women with CPP would perceive the quality of their parental bonding in childhood as more dysfunctional than pain-free women, that anxiety and depression would be associated with dysfunctional parenting styles in women with CPP, and that pain catastrophizing would be associated with greater pain intensity, a higher consumption of pain medication, longer duration of pain, anxiety, depression and adverse parenting styles.

## 2. Materials and Methods

### 2.1. Study Design

An observational, case–control study was conducted between May 2018 and August 2021 at the CPP outpatient clinic and family planning outpatient clinic in the Department of Obstetrics and Gynecology, Teaching Hospital of the Federal University of Goiás, Goiânia, Goiás, Brazil.

### 2.2. Sample Size

The sample size requirements for case–control study designs [33] were taken into consideration. Sample size calculation was based on the following parameters: significance level of 0.05 (α = 0.05), statistical power of 0.80 (β = 0.20), ratio of cases to controls of 1 (k = 1), odds ratio of 2.5 (OR = 2.5), and an expected proportion of dysfunctional parenting style of 54.4% in women with CPP, as previously found in a pilot study [34]. The minimum number of women needed for the total sample was estimated at 202, 101 women with CPP and 101 pain-free controls.

### 2.3. Participants

Women attending the CPP outpatient clinic, or the family planning outpatient clinic (women without CPP/control group) were consecutively invited to participate in the study. The criteria for inclusion in the CPP group were: age ≥18 years and ≤50 years, and having experienced lower abdominal pain during the preceding three months that had begun at least six months previously and was not exclusively cyclical or coitus-related, irrespective of whether the woman was receiving treatment for CPP or the cause of CPP. The criteria for inclusion in the control group were age ≥18 years and ≤50 years and not having experienced lower abdominal pain in the preceding three months. The range of age defined for both groups was ≥18 and ≤50 years, taking into consideration the legal age of consent and the end of the reproductive years, respectively. Pregnancy, either current or in the preceding 12 months, post-menopause, surgery in the previous three months and a history of cancer were considered exclusion criteria for both groups. Providing signed informed consent to participate in the study was part of the inclusion criteria in both groups. Table 1 describes the inclusion and exclusion criteria.

### 2.4. Procedure

Individual interviews were conducted to obtain sociodemographic, behavioral, and clinical data, and specific instruments were used to assess parental bonding, anxiety, and depression. In the group of women with CPP, further investigation was conducted to assess pain catastrophizing. Data collection was performed in a private room to guarantee participants’ privacy.

### 2.5. Measures

#### 2.5.1. Demographic Data

The following sociodemographic, behavioral and clinical characteristics were investigated in both groups: age, skin color (white/non-white), body mass index (BMI; kg/m^2^), years of schooling (<12 years/≥12 years), marital status (partner/no partner), monthly per capita income (Brazilian real, R$), employment status (employed, in paid work/unemployed, retired, homemaker), whether physically active (≥150 min/week) in the preceding month (yes/no), smoker (yes/no), alcohol consumption in the previous three months (yes/no), parity (0/≥1) and a history of abdominal and/or pelvic surgery (yes/no). Women who currently smoked or who had stopped smoking within the previous year were considered smokers, while those who had never smoked or who had stopped smoking more than a year previously were considered non-smokers. Physical abuse was investigated from the question: “Have you ever suffered physical abuse?” (yes/no). Sexual abuse was investigated by asking: “Have you ever suffered sexual abuse?” (yes/no). Difficulty relating to others was identified from answers to the question: “Are you having difficulties in your relationship with someone or are you in conflict with anyone?” (yes/no). A history of chronic disease was investigated based on answers to the following question: “Do you have any chronic disease such as hypertension, diabetes, hypothyroidism or migraine?” (yes/no).

The participants of the CPP group were asked about the duration of their pain. Pain intensity in the two weeks preceding the interview was investigated using a 10 cm (0–10 points) numerical rating scale in which 0 represents the complete absence of pain and 10 the worst pain imaginable. Participants in this group were also asked whether they had used any medication to manage their pain in the preceding month (yes/no).

#### 2.5.2. Assessment of Parental Bonding

The Parental Bonding Instrument (PBI) was used to determine parenting styles as perceived by the participants based on memories of their parents over the first sixteen years of their life [22]. This 25-item questionnaire consists of two domains: care (12 items) and overprotection (13 items). The “care” domain comprises items such as: “Spoke to me in a warm and friendly voice” and “Made me feel I wasn’t wanted”. Items in the “overprotection” domain include: “Invaded my privacy” and “Let me decide things for myself”. Specific scoring instructions for the PBI are based on a 4-point, Likert-type scale (very like, moderately like, moderately unlike and very unlike), with scores that range from 0 to 3; however, not all the items are scored in the same direction [22]. This instrument is completed with regard to both mothers and fathers separately, using separate cut-off points established for each domain. A care score of 27.0 and a protection score of 13.5 for mothers and a care score of 24.0 and a protection score of 12.5 for fathers have been established as the cut-off points for assigning individuals to “high” or “low” categories of parental bonding [22]. The PBI has been validated cross-culturally for use in a variety of languages, including Brazilian Portuguese [35].

#### 2.5.3. Assessment of Anxiety

Symptoms of anxiety were investigated using a version of the Generalized Anxiety Disorder-7 (GAD-7) scale that has been translated into Brazilian Portuguese and validated for use in Brazil [36]. This is a 7-item, self-report questionnaire referring to the preceding two weeks, with questions such as: “How often have you been bothered by becoming easily annoyed or irritated” or “How often have you been bothered by not being able to stop or control worrying”. Scores range from 0 to 3 for each item (0 = not at all; 1 = several days; 2 = more than half the days; and 3 = nearly every day). Total scores range from 0 to 21, with ≥10 being the cut-off score for anxiety [37].

#### 2.5.4. Assessment of Depression

The 9-item Patient Health Questionnaire (PHQ-9), translated and validated for use in Brazilian Portuguese [38], was used to identify symptoms of depression over the preceding two weeks. This self-administered questionnaire is based on the diagnostic criteria for major depression outlined in the Diagnostic and Statistical Manual of Mental Disorders, fourth edition (DSM-IV) and includes the question: “Over the last two weeks, how often have you been bothered by any of the following problems?”, e.g., “feeling down, depressed or hopeless”. Scores range from 0 to 3 for each item, with 0 meaning “not at all”; 1 “several days”; 2 “more than half the days” and 3 “nearly every day”. A score ≥10/27 is considered indicative of depression [37].

#### 2.5.5. Assessment of Pain Catastrophizing

The Pain Catastrophizing Scale (PCS), translated and validated for use in Brazilian Portuguese [39], was used to identify catastrophizing [40] in the CPP group. This self-report questionnaire consists of 13 items distributed across three domains: helplessness, magnification and rumination. The instrument asks patients to reflect on past painful experiences and to indicate the degree to which they experienced each of the 13 thoughts or feelings when experiencing pain, such as: “It’s terrible and I think it’s never going to get any better”. The total score for the PCS ranges from 0 to 52, with scores for each item being based on a 5-point, Likert-type scale ranging from 0 (not at all) to 4 (all the time). A score ≥30 is considered indicative of a clinically relevant level of catastrophizing [41].

### 2.6. Statistical Analysis

In the descriptive analysis, absolute and relative frequencies were calculated for the qualitative or categorical variables, and means and standard deviations (SD) were calculated for the continuous quantitative variables. Yates’s chi-square test and the Mann–Whitney test were used to compare characteristics between the groups. The frequency of parenting styles was compared between the two groups using Yates’s chi-square test. Univariate and multiple logistic regression analyses were conducted to determine possible associations between each parenting style and CPP in women, and the odds ratios (OR) and their respective 95% confidence intervals (CI) were calculated in the initial unadjusted form and then after adjustment for the following potential confounders in addition to parenting style: age, alcohol consumption, physical abuse, sexual abuse, relationship difficulties, abdominal/pelvic surgery, anxiety and depression. Spearman’s correlation coefficient (r) was used to verify possible correlations between catastrophizing, pain intensity, the duration of pain, anxiety, depression, and parental bonding style (maternal and paternal care and overprotection) in the group of women with CPP. The strength of the correlation was described as zero (0), weak (0–0.3), moderate (0.4–0.6), strong (0.7–0.9) and perfect (1), including the respective direction: positive (+) or negative (−) [42]. The SPSS software package, version 20, was used throughout the statistical analysis. Significance level was set at 0.05 [43].

### 2.7. Ethics

All the procedures performed complied with the requirements established in the Declaration of Helsinki. The internal review board of the Federal University of Goiás Teaching Hospital reviewed and approved the study protocol under reference number 2631464. All the participants signed a written informed consent form.

## 3. Results

Overall, 294 women were invited to participate in the study. Since 28 women with CPP and 20 potential controls were excluded, the final sample consisted of 246 women, 123 recruited to the CPP group and 123 to the control group of women without CPP, as shown in Figure 1.

Table 2 describes the characteristics of the participants. The mean age of the women in the CPP group was 37.0 ± 6.9 years (± SD) compared to 31.9 ± 7.2 years for those in the control group (*p* < 0.001). Relationship difficulties were more common in the women with CPP (39.8% in the CPP group vs. 19.5% in the control group; *p* = 0.001). In the CPP group, 79.7% of the women were found to have anxiety compared to 56.9% in the control group (*p* < 0.001), while depression was identified in 73.2% of women in the CPP group and in 56.1% of the controls (*p* = 0.008).

Perceived low maternal care was significantly more common in women with CPP (60.7% vs. 45.2%; *p* = 0.026). There was no statistically significant difference between the groups insofar as the other parental bonding styles were concerned (Table 3). Six women with CPP and eight controls had had no contact with their mothers during childhood and/or adolescence, while 24.8% of the study population (32 women with CPP and 29 controls) reported having had no contact with their fathers during that period (data not shown as table).

In the unadjusted analysis, low maternal care was significantly associated with CPP (OR = 1.87; 95%CI: 1.11–3.15; *p* = 0.019). This association disappeared in the multiple regression analysis following adjustment for potential confounders (Table 4).

In the CPP group, mean values were as follows: duration of pain 7.1 ± 6.5 years, and intensity of pain 7.2 ± 2.4. Catastrophizing was identified in 77.2% of women in the CPP group (data not shown as table). The monthly consumption of pain medication was higher among catastrophizing women compared to non-catastrophizing women (94.7% vs. 78.6%; *p* = 0.024) (data not shown in table). There was a positive and weak correlation between catastrophizing and the intensity of the pain (r = 0.342; *p* < 0.001), anxiety (r = 0.271; *p* = 0.002), depression (r = 0.272; *p* = 0.002) and maternal overprotection (r = 0.185; *p* = 0.046). A negative and weak correlation was found between maternal care and anxiety (r= −0.184; *p* = 0.047), maternal care and depression (r = −0.219; *p* = 0.018), paternal care and anxiety (r = −0.286; *p* = 0.006) and paternal care and depression (r = −0.234; *p* = 0.026). Other correlations are shown in Table 5.

## 4. Discussion

To the best of our knowledge, this is the first study to investigate the association between perceived parenting styles and CPP in women, and the relationship between catastrophizing and parenting styles in women with this condition. In the current study, low maternal care was more common in women with CPP. In addition, low maternal care was significantly associated with CPP; however, following adjustment for possible confounders, this association was no longer present. Lower maternal and paternal care was also associated with higher anxiety and depression in women with CPP. Catastrophizing correlated positively and weakly with maternal overprotection, the intensity of pain, anxiety, and depression in women with CPP.

The frequency of low maternal care was significantly higher in women with CPP compared to that found in the control group. Previously, a pilot study with a small sample size and without a control group had identified 68.2% of low maternal care in women with CPP [34]. This percentage is slightly higher than that found in the current study (60.7%). Furthermore, the association that was initially found between low maternal care and CPP in women in the unadjusted analysis corroborates other reports involving parenting styles and other types of persistent pain. Indeed, lack of maternal sensitivity in childhood has already been associated with intense chronic pain in adulthood [28] and an association has also been found between low maternal care and chronic pain in adolescents [26]. In addition to the association with the development of chronic pain, parental low care was found to be associated with the patient’s need for psychosomatic treatment in adulthood [24]. This suggests that maternal care in childhood may have an effect on individuals’ experiences of pain at different stages of their lives.

Nevertheless, in the current study, the association between low maternal care and CPP disappeared following adjustment for potential confounders. Likewise, a previous study reported that an association between low maternal care in childhood and chronic pain in a general adult population was no longer present following adjustment for depression [27]. In addition, other studies have shown the role of depression as a mediator in the association between low maternal care and chronic pain in adolescents [26] and in adults [28]. In this respect, the impact of the quality of maternal care in childhood on the structural and functional development of parts of the nervous system linked to emotional regulation was demonstrated in a study conducted using magnetic resonance imaging [44]. A better quality of maternal care was associated with a greater volume and greater activation of the middle frontal cortex [44], an important structure in the top-down control over amygdala reactivity to stressors [45]. Conversely, poorer quality of maternal care was associated with a decreased volume and increased activation of the hippocampus [44], with a long-term effect on the negative feedback system of the hypothalamic-pituitary-adrenal (HPA) axis stress response [46].

A negative association was found between maternal care and anxiety and depression in the group of women with CPP in the present study. Notwithstanding, although significant, this association was weak. As already reported in other studies [47], these emotional disorders were significantly more common in women with CPP compared to pain-free control. There is increasingly robust evidence in the literature of a bidirectional association between anxiety, depression and chronic pain through shared neural circuits [48,49] and neurotransmitters [50,51] in the expression of these conditions. Since the relationship between children and their mothers is of a biological and deeply emotional nature, the quality of this bond may be affected by sociocultural changes in the women’s role within their families. Therefore, maternal care in childhood, as well as the different family compositions and the roles of each member of the family, merit further investigation regarding their possible direct or indirect effects on CPP in women.

No association was found in the present study between paternal bonding styles in childhood and CPP in women. Conversely, in other clinical settings, some authors have found higher paternal overprotection in childhood to be associated with other types of chronic pain in adolescents [26] and in adults [24,27]. Paternal bonding is predominantly shaped by sociocultural aspects, which may explain, at least in part, the differences between the findings of the present study and those of other authors whose studies involved fathers of other nationalities. Interestingly, in the present study, a negative correlation was found between paternal care and anxiety and depression in women with CPP. This finding is in agreement with the results of a recent study that reported an effect of paternal as well as maternal parenting on the maturation of the HPA axis and the child’s immune system [11]. This may be due to the increasingly common transformations in the father’s role as caregiver in modern society, and merits further investigation in future studies. In the present study, however, 24.8% of the sample population had had no contact with their father for a variety of different reasons, which may have affected the results obtained in relation to paternal bonding and its possible association with CPP in women.

A high frequency of catastrophizing (77.2%) was found in the group of women with CPP. Albeit lower, a considerable frequency of catastrophizing (53.1%) was found in another study involving women with CPP [30]. In the present study, catastrophizing women were found to use significantly more pain medication compared to non-catastrophizing women, highlighting the clinical and economic impact of this coping strategy. The present results showed a positive and significant association between catastrophizing and pain intensity, anxiety, and depression. Other authors have reported not only a greater intensity of pain [30] but also pain hypersensitivity [12], anxiety, depression, and poorer quality of life associated with catastrophizing in women with CPP [30]. Furthermore, outcomes were found to be poor in catastrophizing women with CPP associated with endometriosis [52] or other causes [31], thus highlighting the relevance of including catastrophizing in the approach used to treat this health condition. It is important to mention that the association found between catastrophizing and pain intensity, anxiety and depression, albeit significant, was weak, and this has to be taken into account when considering the clinical and social interpretation of these findings. The presence of other caregivers and their role in the experience of pain experienced by women with CPP were not investigated further in this study and could have affected its results.

The finding of a positive and weak, although significant, correlation between maternal overprotection and catastrophizing in the women with CPP in the present study is also noteworthy. Since, to the best of our knowledge, this finding is novel, comparison with other studies had necessarily to involve research into other types of pain. A previous study found that maternal overprotection acted as a mediator in the catastrophizing responses of children with functional abdominal pain [10]. Conversely, an unpredictable maternal care model has been associated with an insecure attachment type and migraine in children and adolescents [9,53] and migraine is a common comorbidity in women with CPP [54]. Stressful parent bonding, particularly maternal dominance, has been associated with an increase in the functional and emotional disturbances of primary dysmenorrhea [55]. Results from the Tracking Adolescents’ Individual Lives Survey (TRAILS), a prospective cohort study involving 2230 adolescents in the Netherlands, are also of interest [56]. In that study, parental overprotection was found to be predictive of the development of functional somatic symptoms such as aches/pain, with this association being stronger in relation to the maternal overprotection of daughters [56]. Furthermore, a controlling family environment contributes to poorer long term psychosocial functioning in young adults with fibromyalgia [57], a condition that often overlaps CPP in women [54]. Although each one of these studies evaluated different types of pain in different populations, their findings and those of the present study appear to be aligned. In-depth research into the role of families, particularly parents, in the emotional development and in the development of pain coping strategies in children may have a preventive and therapeutic effect on CPP in women.

Furthermore, the history of the participants in the present study suggests a significantly higher frequency of physical abuse and relationship difficulties in the women in the CPP group compared to the controls. In agreement with the present findings, women with CPP have been described as having a history significantly associated with ACEs, particularly physical abuse, sexual abuse and emotional abuse, and of having witnessed domestic violence [32], with a high rate of co-occurrence of anxiety and depression [32,47]. Therefore, irrespective of a cause-effect relationship, these issues must be properly addressed in women with CPP.

The importance of an interdisciplinary approach involving psychosocial aspects in the treatment of CPP in women has already been well-established [7]. Mindfulness-based stress reduction and cognitive behavioral therapy have proven useful in reducing pain catastrophizing [31]. The present findings support the need for further, more specific research that would produce robust evidence on the role of the early parent–child relationship in the development of emotional self-regulation and coping skills in women with CPP. Therefore, therapeutic approaches with an emphasis on redefining family history and relationships, including family constellation [58], as well as other forms of therapy that encourage experiences of self-discovery and self-regulation such as Identity-Oriented Psychotrauma Therapy [59] could perhaps be included in studies into this intriguing health condition that is CPP in women.

Finally, the management of CPP continues to represent a challenge to healthcare professionals and patients. This study could serve as a starting point for future research into other aspects of interpersonal relationships, parenting and attachment styles, and CPP in women. This could lead to improvements in the prevention and treatment of this important health condition.

### Strengths and Limitations

Some limitations need to be taken into consideration when interpreting the findings of the present study. The use of self-report questionnaires may have generated subjectivity in the responses obtained. A history of physical abuse, sexual abuse and relationship difficulties was investigated through single questions requiring yes/no answers. Validated questionnaires on these topics were not used. Only maternal and paternal bonding was considered objects of interest, with the role of other figures of affection when the parents were unavailable not being investigated. Moreover, a quarter of the population evaluated here had had no contact with their father; therefore, this finding cannot be extrapolated. Attachment styles, i.e., the patterns of interpersonal relationships in adulthood based on early life experiences [23], were not evaluated in the present study and should be the focus of future studies. Although there was a difference in the frequency of anxiety, depression and physical abuse between the groups, the analysis was adjusted for these variables, which may have minimized their impact on the results.

Strongpoints include the fact that all the instruments used here have been transculturally validated for use in Brazilian Portuguese. Although the PBI accesses information from memories of the individual’s perceptions on their mother or father in childhood, the stability and consistency of this instrument have been validated over the long-term [60]. In addition, the fact that this was one of the largest studies in terms of sample size to evaluate women with CPP is also worthy of mention.

## 5. Conclusions

The frequency of low maternal care was found to be greater in the CPP group compared to the pain-free controls. The domains of parental bonding were not independently associated with CPP, while maternal and paternal care was negatively associated with anxiety and depression in women with CPP. Catastrophizing was very common and positively associated with the intensity of pain, anxiety, depression, and maternal overprotection in the CPP group. These associations, albeit significant, were weak. The findings of the current study should be interpreted taking the psychosocial context of the study population into consideration. Furthermore, these data should be taken into account when planning future studies on the subject and particularly with respect to the clinical management of women with CPP.

## Figures and Tables

**Figure 1 ijerph-19-13347-f001:**
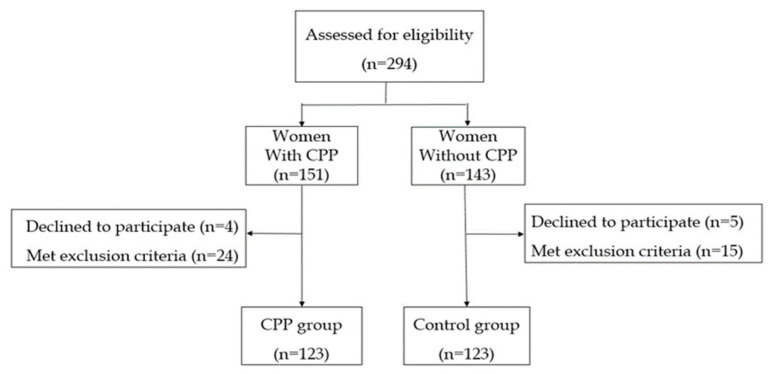
Flowchart of the study population. CPP: Chronic Pelvic Pain.

**Table 1 ijerph-19-13347-t001:** Inclusion and exclusion criteria.

	Inclusion Criteria	Exclusion Criteria
Both groups	Age ≥18 and ≤50 years Signed informed consent	Being currently pregnant or having been pregnant in the preceding 12 months Being postmenopausal Having undergone any surgery in the previous 3 months History of cancer
CPP group	Lower abdominal pain for ≥6 months, not exclusively cyclical or coitus-related	
Pain-free group	No lower abdominal pain in the preceding 3 months	

CPP: chronic pelvic pain.

**Table 2 ijerph-19-13347-t002:** Characteristics of the study participants.

Characteristics	CPP Group (n = 123)	Control Group (n = 123)	*p*-Value
Age (mean ± SD; years) ^1^	37.0 ± 6.9	31.9 ± 7.2	<0.001 ^c^
Body mass index (mean ± SD; kg/m^2^) ^1^	27.5 ± 5.4	27.1 ± 5.8	0.310
Monthly per capita income (mean ± SD; R$) ^1^	743.28 ± 496.68	748.08 ± 645.27	0.926
	n (%)	n (%)	
Skin color ^2^			0.447
White	31 (25.2)	25 (20.3)	
Non-white	92 (74.8)	98 (79.7)	
Years of schooling ^2^			0.347
<12 years	46 (37.4)	38 (30.9)	
≥12 years	77 (62.6)	85 (69.1)	
Employment status ^2^			0.797
Employed/in paid work	67 (54.5)	70 (56.9)	
Unemployed/retired/homemaker	56 (45.5)	53 (43.1)	
Marital status ^2^			0.893
With a partner	82 (66.7)	80 (65.0)	
No partner	41 (33.3)	43 (35.0)	
Smoking ^2^			0.055
Smoker	10 (8.1)	21 (17.1)	
Non-smoker	113 (91.9)	102 (82.9)	
Alcohol consumption ^2^			0.002 ^b^
Yes	48 (39.0)	73 (59.3)	
No	75 (61.0)	50 (40.7)	
Physically active ^2^			0.517
Yes	26 (21.1)	21 (17.1)	
No	97 (78.9)	102 (82.9)	
Physical abuse ^2^			0.003 ^b^
Yes	39 (31.7)	18 (14.6)	
No	84 (68.3)	105 (85.4)	
Sexual abuse ^2^			0.062
Yes	27 (22.0)	15 (12.2)	
No	96 (78.0)	108 (87.8)	
Relationship difficulties ^2^			0.001 ^b^
Yes	49 (39.8)	24 (19.5)	
No	74 (60.2)	99 (80.5)	
Parity ^2^			0.148
0	29 (23.6)	19 (15.4)	
≥1	94 (76.4)	104 (84.6)	
Chronic disease ^2^			0.071
Yes	93 (75.6)	79 (64.2)	
No	30 (24.4)	44 (35.8)	
Abdominal/pelvic surgery ^2^			0.017 ^a^
Yes	96 (78.0)	78 (63.4)	
No	27 (22.0)	45 (36.6)	
Anxiety ^2^			<0.001 ^c^
Yes	98 (79.7)	70 (56.9)	
No	25 (20.3)	53 (43.1)	
Depression ^2^			0.008 ^b^
Yes	90 (73.2)	69 (56.1)	
No	33 (26.8)	54 (43.9)	

CPP: Chronic pelvic pain. SD: standard deviation. ^1^ Mann–Whitney U test. ^2^ Yates chi-square test. ^a^
*p* < 0.05; ^b^
*p* < 0.01; ^c^
*p* < 0.001.

**Table 3 ijerph-19-13347-t003:** Parenting styles in women with chronic pelvic pain (CPP) compared to pain-free controls.

Parenting Styles	CPP Group		Pain-Free Controls		*p*-Value
	n	%	n	%	
Maternal	117		115		
Care					
Low	71	60.7	52	45.2	0.026 ^a^
High	46	39.3	63	54.8	
Overprotection					
Low	35	29.9	34	29.6	>0.999
High	82	70.1	81	70.4	
Paternal	91		94		
Care					
Low	45	49.5	46	48.9	>0.999
High	46	50.5	48	51.1	
Overprotection					
Low	22	24.2	30	31.9	0.314
High	69	75.8	64	68.1	

Yates chi-Square test. ^a^
*p* < 0.05.

**Table 4 ijerph-19-13347-t004:** Association between parenting styles and chronic pelvic pain in women.

Parenting Styles	Chronic Pelvic Pain			
	Unadjusted		Adjusted ^1^	
	OR (95%CI)	*p*-Value	OR (95%CI)	*p*-Value
Maternal				
Care				
High	1.0 (reference)		1.0 (reference)	
Low	1.87 (1.11–3.15)	0.019 ^a^	1.38 (0.74–2.57)	0.315
Overprotection				
Low	1.0 (reference)		1.0 (reference)	
High	0.98 (0.56–1.73)	0.954	0.65 (0.33–1.28)	0.212
Paternal				
Care				
High	1.0 (reference)		1.0 (reference)	
Low	1.02 (0.57–1.82)	0.944	0.91 (0.46–1.81)	0.799
Overprotection				
Low	1.0 (reference)		1.0 (reference)	
High	1.47 (0.77–2.81)	0.243	1.12 (0.51–2.43)	0.782

OR: odds ratio. CI: confidence interval. ^1^ Multiple logistic regression adjusted for age, alcohol consumption, physical abuse, sexual abuse, relationship difficulties, abdominal/pelvic surgery, anxiety, and depression. ^a^
*p* < 0.05.

**Table 5 ijerph-19-13347-t005:** Spearman correlation between characteristics evaluated in women with chronic pelvic pain.

	Pain Intensity	Duration of Pain	Catastrophizing	Anxiety	Depression	Maternal Care	Maternal Overprotection	Paternal Care	Paternal Overprotection
Pain intensity	1.0	−0.051	0.342 ^c^	0.208 ^a^	0.165	−0.101	−0.098	0.087	−0.153
Duration of pain		1.0	0.028	−0.172	−0.130	0.040	−0.157	0.080	−0.058
Catastrophizing			1.0	0.271 ^b^	0.272 ^b^	−0.128	0.185 ^a^	−0.175	0.102
Anxiety				1.0	0.717 ^c^	−0.184 ^a^	0.025	−0.286 ^b^	−0.003
Depression					1.0	−0.219 ^a^	−0.040	−0.234 ^a^	−0.003
Maternal care						1.0	−0.183 ^a^	0.224 ^a^	0.140
Maternal overprotection							1.0	−0.146	0.368 ^c^
Paternal care								1.0	−0.079
Paternal overprotection									1.0

^a^*p* < 0.05; ^b^
*p* < 0.01; ^c^
*p* < 0.001.

## Data Availability

All data generated or analyzed during this study are included in this published article.

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
