# Peer review of "Parenting Styles, Mental Health, and Catastrophizing in Women with Chronic Pelvic Pain: A Case-Control Study"

_ijerph, 2022, doi:10.3390/ijerph192013347_

Round 1

Reviewer 1 Report

Congratulations on the study. I am suggesting only minor corrections and the inclusion of two paragraphs to increase the relevance of the study. Please, see the attached file,

Reviewer 2 Report

The research is in an important area. The theoretical and practical reasoning behind conducting this study needs is strong in the introduction and outlined in detail in the discussion. The references are rich and without excessive self-citations. The methodology is presented clearly and does need any more comments. I wish that all the hypotheses set by the authors will be listed and commented on in the discussion. My proposition is to present the inclusion and exclusion criteria in the table. In the section on inclusion criteria - informed consent should be mentioned. The results are presented well, and the broad discussion is impressive and sufficient. In tables all the acronyms and abbreviations should be listed and explained under the table for example - SD - standard deviation, etc.

Reviewer 3 Report

Parenting styles, mental health, and catastrophizing in women with chronic pelvic pain: a case-control study

This is a very interesting paper and I congratulate the authors on the job welldone. A couple of things to thing about.

It seems plausible that one would expect higher levels of anxiety, depression and physical violence in the case arm as opposed to the controls, literature demonstrates a multi-dimensional impact of pain. How is this accounted for? One would expect a detailed examination of the limitations of a case control study to be discussed in this situation.

Were these data examined for the nature of distribution? How were missing values and outliers handled?

Some details regarding how some of the variables were measured. Forexample physical abuse, which was assessed using just one global question. Then sexual abuse ( do you assume all respondents know about the scope of what defines sexual abuse?)  questions like are you in conflict with anyone- what does this mean to an average woman? I think this should be acknowledged in the limitations section.

 The authors report positive correlations CPP and pain intensity, anxiety, depression etc but the shared variance is very low, even the correlations are low. It would be useful to include the various cut offs and for the different strengths of correlation i.e low, moderate, high etc so that readers have a sense of what makes clinical or social meaning in terms strength of association or shared variance.

In the discussion section, I guess what makes sense is to explain more the meaning of the finding that low maternal care was mot associated with CPP. It simply wasn’t and the decision-making result is that which controls for confounders.

The correlations are also low, and the shared variance is low, again the authors should focus on explaining these findings as opposed to emphasizing existence of a positive correlation.
